**∂ | Open Peer Review** | Bacteriology | Research Article

# Profiling of *Burkholderia pseudomallei* variants derived from Queensland's clinical isolates

Pauline M. L. Coulon,[1] Kay Ramsay,[2] Aven Lee,[3] Edita Ritmejeryte,[3] Miranda E. Pitt,[1] Joyce To,[1] Daniel G. Mediati,[1] Ian Gassiep,[2] Sarah Reed,[3] Patrick N. A. Harris,[2] Garry S. A. Myers[1]

**ABSTRACT** *Burkholderia pseudomallei* (*Bp*), an environmental bacterium and opportunistic pathogen endemic to tropical regions, is highly adaptive and thrives in diverse environments, from soil to human hosts. Bacterial adaptation is critical for survival, virulence modulation, and persistence during infection and can manifest as colony morphotype variation (CMV). Although *Bp* adaptation has been studied, CMV remains poorly understood. Here, we characterized five clinical *Bp* isolates exhibiting heterogeneous populations with rough and smooth colony morphologies. We used phenotypic assays, whole-genome sequencing, and proteomics to investigate the molecular pathways reflecting CMV, by comparing smooth and rough morphotypes. Although phenotypic differences in protease activity, hemolysis, mucoidy, iron uptake, and antibiotic sensitivity—including to antimicrobial agents commonly used to treat infections—were rare, these traits alone could not distinguish morphotypes or groups of isolates. Genomic comparisons revealed either no differences or limited isolate-specific mutations, which do not explain the overall difference in phenotypes. In contrast, proteomic analysis uncovered consistent shifts in protein abundance related to virulence, including quorum sensing, DNA methylation, and secretion systems. Rough variants showed higher abundance of EPS-associated proteins, the BpsI3/R3 quorum-sensing system, and the global regulator ScmR, whereas smooth variants displayed higher abundances of proteins belonging to type III/VI secretion and siderophore biosynthesis pathways. These findings suggest that CMV is driven by phase variation and regulatory mechanisms rather than punctual genomic modifications. Our study underscores the limitations of phenotype or genome-based classification alone in the context of CMV and highlights the value of integrated multi-omics approaches to uncover CMV-associated biomarkers, with potential applications in diagnostics and the development of targeted therapies against persistent and drug-resistant *Bp* infections.

**IMPORTANCE** *Burkholderia pseudomallei* (*Bp*), the causative agent of melioidosis, is endemic to Australia, Asia, Africa, and the Americas. It predominantly affects Indigenous populations and individuals suffering from diabetes, chronic lung or kidney disease, or alcoholism. *Bp* is known for its exceptional genomic and phenotypic plasticity, enabling rapid adaptation to diverse environments. This adaptability is reflected by colony morphotype variation (CMV), including reversible phase variation between smooth and rough colonies. In this study, we report rough and smooth colonies from clinical samples and emphasize the importance of characterizing CMV through multi-omics approaches rather than relying solely on genomics and phenotypic traits. By integrating genomic, phenotypic, and proteomic data, we identified that a limited number of mutations, including one in a putative regulatory element, likely drive major molecular changes between morphotypes. These affect the expression of quorum-sensing systems, the transcriptional regulator ScmR, DNA methyltransferase, and virulence-associated genes.

Address correspondence to Pauline M. L. Coulon, pauline.coulon@uts.edu.au.

The authors declare no conflict of interest.

**KEYWORDS** colony morphotype variation, epigenetics, quorum sensing, omics, phase variation

*B*urkholderia are highly adaptable gram-negative environmental saprophyte bacteria found in diverse ecological niches ranging from soil and plant roots to human hosts. Their remarkable phenotypic plasticity enables them to persist in various ecological niches, often facilitated by transmission through contaminated water, soil, and products (1–7). Among them, *Burkholderia pseudomallei* (*Bp*) poses a significant public health threat as the causative agent of melioidosis, a severe and often fatal disease in animals and humans (1, 8). Although *Bp* is endemic to tropical and sub-tropical regions of Asia, Africa, Central and South America, and Australia (1), it has been recognized since 2023 as naturally established in the environment of the United States (9).

While most healthy individuals exposed to *Bp* do not develop melioidosis, vulnerable populations living with underlying conditions such as cystic fibrosis, diabetes, chronic kidney disease, chronic lung disease, alcoholism, or a compromised immune system often are the most at risk. In these individuals, *Bp* infections can cause skin abscesses or severe pneumonia, often leading to fatal sepsis with a mortality rate reaching up to 52% (1, 10–17). The incidence of *Bp* infections is strongly correlated with heavy rains and flooding, and cases are expected to increase as climate change intensifies extreme weather events (18). During 2025, Queensland (Australia) reports its worst year of melioidosis in three decades, with over 249 cases and 36 deaths (Queensland Health). Additionally, there is a growing concern that melioidosis cases could (i) spread into New South Wales (Australia), where the disease is less recognized by healthcare providers, and (ii) be transmitted directly by pets, livestock, or wild animals through contact with infected wounds or via shared environment—even if no zoonotic transmission has been confirmed to date (8).

*Burkholderia*'s rapid adaptation to environmental changes and stressors is often associated with modifications in outer membrane proteins, antibiotic resistance, and virulence factors (19–23). This adaptability is reflected by colony morphotype variation (CMV), a process by which bacteria switch between distinct colony morphologies. CMV is now used as an umbrella term encompassing reversible changes (phase variation without genomic plasticity), irreversible genomic plasticity (e.g., chromosome loss (24)), antigenic variation, and phenotypic switching (25, 26). Although CMV plays a critical role in persistence during infection, it remains underexplored. However, a few studies have shown that CMV can result in mutations in global regulators, two-component systems, genome reduction and duplication, bacteriophage cluster integration, and DNA methylation, as reviewed by Coulon and colleagues (26).

Among *Burkholderia* species, *Bp* has been the most extensively studied in terms of CMV. Chantratita and colleagues identified up to seven dynamic and reversible colony morphologies in *Bp* (19). Under laboratory conditions, these variants emerge in response to stress conditions and revert to the parental morphotype once the stress is removed (19, 23, 27). Notably, CMVs also occur during infection: in 10% out of over 450 clinical *Bp* isolates, a mixed mucoid and non-mucoid CMV population was observed on blood agar. The mucoidy phenotype was attributed to a difference in O-antigen Lipopolysaccharides (LPS; [OPS]) production, whereas no mutations or altered expression of the *wbiA* O-antigen acetylase was linked (28). Furthermore, the emergence of small-colony variants (SCVs) from a wrinkled *Bp* isolate during long-term infections in murine was attributed to upregulation of the LPS biosynthesis cluster genes (29, 30). Using experimental evolution in the same model, Gierok and colleagues have demonstrated the emergence of two reversible yellow colony variants (YA and YB) from the environmental *Bp* K96243. These variants showed upregulation of the σ–54 dependent regulator *yelR,* leading to resistance to hypoxic stress and enhanced colonization and persistence in the murine stomach (31).

In this study, we characterized five additional *Bp* isolates undergoing CMV using phenotypic, genomic, and proteomic profiling to uncover the similarities and dissimilarities in molecular pathways resulting in CMV.

## MATERIALS AND METHOD

### Colony morphology screening

In brief, 292 *Bp* isolates were collected from Townsville between 1999 and 2022 (32). Among those classified as "mixed population" (Etipola and colleagues, unpublished), five isolates were selected for this study. Among the isolates listed in Table S1, each was streaked on Luria Broth (LB) agar plates containing 0.01% Congo red (CRLA) from −80°C stock and incubated at 37°C for three overnights to confirm the mixed population phenotype.

### Phenotypic assays

#### Production of virulence factors

Individual colonies (four biological replicates) were picked and stabbed into LB agar plate containing 1.5% skimmed milk to assess proteolytic activity (33); 0.5 g/L yeast extract supplemented with 4 g/L D-mannitol (YEM) and agar to assess mucoidy (34); a modified recipe of Chrome Azurol S agar (using 10× chelated LB instead of minimal media nine supplemented with chelated casamino acid) to assess siderophore production (35) and ensure the growth of clinical isolates and Columbian horse blood agar plate to assess hemolysis activity (36). Plates were incubated at 37°C for 3 days. Inhibition circles were measured and confirmed by statistical analyses.

#### Antimicrobial sensitivity assay

Individual colonies were resuspended in 0.9% NaCl solution to achieve a turbidity of 0.5–0.63 McFarland. Each bacterial suspension was then spread on Muller-Hinton II agar plates to assess susceptibility to Meropenem using gradient MIC test strips (ETEST, range: 0.002–32 µg/mL; Biomérieux), and Oxoid disk diffusion for Ceftazidime (10 µg) and Sulfamethoxazole-Trimethoprim (25 µg). Antimicrobial susceptibility was determined according to the EUCAST guidelines (v15). For CMVs from the same isolate showing considerable visual changes in antibiotic susceptibility, three biological replicates were conducted on independent days, and the average inhibition zone diameters were used for susceptibility classification and confirmed by statistical analyses.

#### Statistical analyses

All data sets were first tested for normality using the Shapiro–Wilk test. Based on the results, comparisons of virulence factor production and resistance to antibiotics between rough and smooth CMVs were performed using either a Wilcoxon rank-sum test or a *t*-test, with a Bonferroni-adjusted *P* value.

### Genomic analysis

#### gDNA extraction

Genomic DNA was extracted from a mix of subsamples of three macrocolonies grown for three nights at 37°C on CRLA using either gDNA spin or high molecular weight extraction kits (NEB). Short-read sequencing (Illumina) was conducted on all CMVs. Briefly, libraries were prepared using 1 ng of gDNA via the Nextera XT kit and sequenced using the NovaSeq XPlus platform (37). On select isolates, long-read sequencing (Oxford Nanopore Technologies) was implemented. Libraries were prepared using 400 ng gDNA with the SQK-NBD114.24 kit and sequenced using R10.4.1 flow cells on the GridION platform. Reads were base-called using Dorado 7.6.7.

### Genome assembly

Nanopore and Illumina reads were quality-filtered and trimmed using Nanofilt (v.2.8.0 [38]) and fastp (v0.24.1 [39]), respectively. Nanopore reads were assembled using Autocycler (v0.4.0 [40]). Assemblies were polished using Polypolish (v0.6.0 [41]), and then POLCA from the pypolca toolkit (v3.0.1 [42]) with Illumina reads. Final assemblies were annotated using Bakta (v1.11 [43]). For the second CMV from each isolate, Illumina reads were assembled into contigs using Unicycler (short read option, v.0.5.1 [44]).

### Genomic modification analyses

To identify SNPs and indels, Illumina reads from the second CMV were mapped onto the corresponding first CMV assembly using Snippy (v4.6.0; - Galaxy implementation) (45). As a control, Illumina reads from the first CMV were also mapped onto their own assemblies, confirming the absence of variation.

### Prediction of regulatory targets in non-coding regions

At least 50-nt of the upstream and downstream sequences of non-coding genomic variants (including SNPs, insertions, or deletions) were extracted and used as input into IntaRNA (46) to predict regulatory targets in *Bp*. RNA-RNA interactions were filtered for those occurring within 50-nt upstream and 25-nt downstream of mRNA start codons (AUG), at least 7 nucleotides of base-pairing, and containing no G-U base-pairs at the 5′ or 3′ seed ends.

### Prediction of secondary structure in non-coding regions

The non-coding genomic variant regions (50-nt upstream and downstream, as *above*) were extracted and used as input into the RNAfold software package to calculate the minimum free energy secondary structure formation of sequences (47). The analyses were performed using the -p, -d2, and --noLP default parameters.

### Identification of genes encoding for DNA methyltransferase and their linked motifs

Pod5 files obtained from Nanopore sequencing were merged, using the pod5 merge function, and analyzed for DNA methylation motifs using the Dorado with both dna_r10.4.1_e8.2_400bps_hac@v4.3.0_6mA@v2, dna_r10.4.1_e8.2_400bps_hac@v4.3.0_5mC_5hmC@v1 models (https://github.com/nanoporetech/dorado). To identify the location and number of detected motifs, as well as the genes encoding DNA methyltransferases, the call_methylation and annotate_rm functions from MicrobeMod were used. These tools were provided with the corresponding genome assembly ".fasta" and annotation ".gff3" files for each isolate (Tables S3 and S4).

## Proteomic analysis

### Protein extraction and peptide preparation

After 3 days on CRLA incubated at 37℃, bacterial colonies were resuspended in lysing buffer (1M Tris-HCL containing 1% SDS) and were boiled at 95℃ and cooled down 10 min at 4℃, thrice. Bacterial lysates were then sonicated in an ultrasonic bath YJ5120-1 (Labtex) for 3 min, followed by shaking at 1,000 rpm at 95℃ for 15 min on a Thermomixer Compact (Eppendorf). The mixture was allowed to cool down to room temperature, and 30 µL was mixed with 270 µL of Buffer I (6 M guanidine chloride, 50 mM Tris pH 8.0, 10 mM DTT) and incubated at 30℃ for 40 min. Following the addition of 9 µL of 0.5 M iodoacetamide, the sample was incubated at 30℃ for 50 min. The reduced and alkylated protein sample was transferred to a 10 kDa cut-off Amicon Ultra device (Merck), which was placed into a collection tube and centrifuged at $14,000 \times g$ for 30 min.

The flow-through from the column was removed, and 300 µL of 100 mM ammonium bicarbonate was added to the column, followed by centrifugation at 14,000 × $g$ for 30 min. To digest proteins in the column, 130 µL of 100 mM ammonium bicarbonate containing 2 µg Trypsin was added. The column was incubated at 37°C for 24 h and then centrifuged in a new collection tube at 14,000 × $g$ for 30 min. Fifty microliters of water with 0.1% formic acid was then added to the spin column, followed by centrifugation at 14,000 × $g$ for 30 min to elute the remaining digested peptides. Enzymes and salts were removed using $C_{18}$ ZipTips (Merck) following the manufacturer's instructions, and desalted peptides were eluted with 80% acetonitrile/0.1% formic acid into microcentrifuge tubes. Digested peptide samples were dried down using a vacuum concentrator (Concentrator plus, Eppendorf) and reconstituted with 50 µL water with 0.1% formic acid prior to LC-MS/MS analysis.

### LC-MS/MS analysis

The samples were chromatographically separated and the resulting proteomics data were acquired using an Acquity M-class micro-LC system (Waters, USA) coupled with a ZenoTOF 7600 LC-MS/MS system using the OptiFlow Turbo V ion source (AB Sciex). Two microliters of sample were loaded onto Micro TRAP C18 column (0.3 × 10 mm; Phenomenex) and washed for 10 min at 10 µL/min with 97% solvent A (water with 0.1% [vol/vol] formic acid) and 3% solvent B (99.9% [vol/vol] acetonitrile, 0.1% [vol/vol] formic acid). Liquid chromatography was performed at 5 µL/min using a 2.7 µm Peptide C18 160 Å; (0.3 × 150 mm; Bioshell) column, with the column oven temperature maintained at 40°C. The gradient started at 3% solvent B, which was increased to 5% by 0.5 min, followed by an increase to 35% solvent B by 38.9 min and then another increase to 80% solvent B by 39 min. The column was washed with 80% solvent B for 3.9 min before equilibrating with 3% solvent B from 43 to 45 min. Data were acquired using Zeno SWATH DIA, using a 65 variable width SWATH windows. The TOF MS parameters were as follows: precursor mass range was 400–1,500 m/z; declustering potential (DP) was set to 80 V, and accumulation time was set at 0.25 s. The TOF MS/MS parameters were as follows: fragmentation mass range was set to 100–1,500 m/z, with an accumulation time of 0.025 s; fragmentation mode was set to CID with Zeno pulsing selected. The collision energy for the MS/MS acquisition was automatically adjusted according to the m/z and charge of the peptide.

### Data analyses

Raw label-free quantification (LFQ) LC-MS/MS data were converted to .mzML files, using MS convert, to allow the data search using Fragpipe v22 (48). Protein identification was conducted against the respective annotated proteomes using the DIA_SpecLib_Quant workflow, which builds a spectral library with MSFragger-DIA and performs quantification with DIA-NN. The output "diann-output.pg_matrix.tsv" was filtered to remove contaminants and retain proteins with at least two non-missing values for at least one group. Data were normalized using the variance stabilization normalization (49). Proteins with two or more missing values in one group were categorized as "detected/not detected." For proteins with a single missing value in one group, missing data were imputed using the mean of the other replicates in that group. Differential protein expression between groups was assessed using Welch's $t$-test, with Benjamini–Hochberg correction applied to control the false discovery rate (adjusted $P < 0.05$) (50, 51). Analyses were performed on three biological replicates (Tables S5 to S9).

### Proteomic representation

All differences in protein abundances observed among the Townsville $Bp$ isolates in this study were mapped back to the reference strain $Bp$ K96243, widely regarded as the model organism for $Bp$ research and the basis for most virulence factor and pathway annotations. To identify homologous proteins between the Townsville $Bp$ isolates and $Bp$

K96243, a megablastp was set up with only one output match for each gene in query using the following parameters: -evalue 10 -outfmt 6 -num_threads 3 -max_target_seqs 1 -max_hsps 1 (Table S10) (52). Clustering of orthologous groups (COGs) for annotated proteins was determined using the EggNOG tool v5 (Table S11) (53). Heatmap representation, made using "pheatmap" package in R (54), was based on protein with at least two isolates showing differences with either three not detected by LC-MS/MS or with a maximum of two not showing differential abundance between CMV.

## Multi-omics integration

### Principal component analysis

Data were transformed using a binary variable to be able to create a principal component analysis using "stats" R package function "prcomp" (55). PCA was conducted separately for each omics layer using, respectively, 16 phenotypes, 19 genomic variables, and the top 30 proteomic variables. For integrative analysis, all genomic and phenotypic variables, along with the top 30 most variable proteomic features, were combined to explore multi-omics clustering patterns across isolates and colony morphotypes.

## RESULTS

### Colony morphotype variation does not translate into distinct phenotypic clusters

Among over 200 clinical *Bp* isolates from their Townsville collection (32), Etipola and colleagues (unpublished) classified them into wrinkled, mucoid, or mixed colony morphotypes when grown on Ashdown selective agar. From the group classified as having "mixed" morphologies, we selected seven wrinkled clinical isolates and their corresponding mucoid variants for further genomic, proteomic, and phenotypic investigations (Table S1). Since CMV can occur spontaneously under laboratory conditions, such as removing antibiotic pressure (19, 23, 27), we streaked each isolate onto CRLA to confirm its capacity to generate variants.

Isolates TSV267 and TSV274 produced only one type of colony morphology on CRLA. Therefore, the five remaining isolates (TSV82, 140, 268, 270, and 287) were selected for

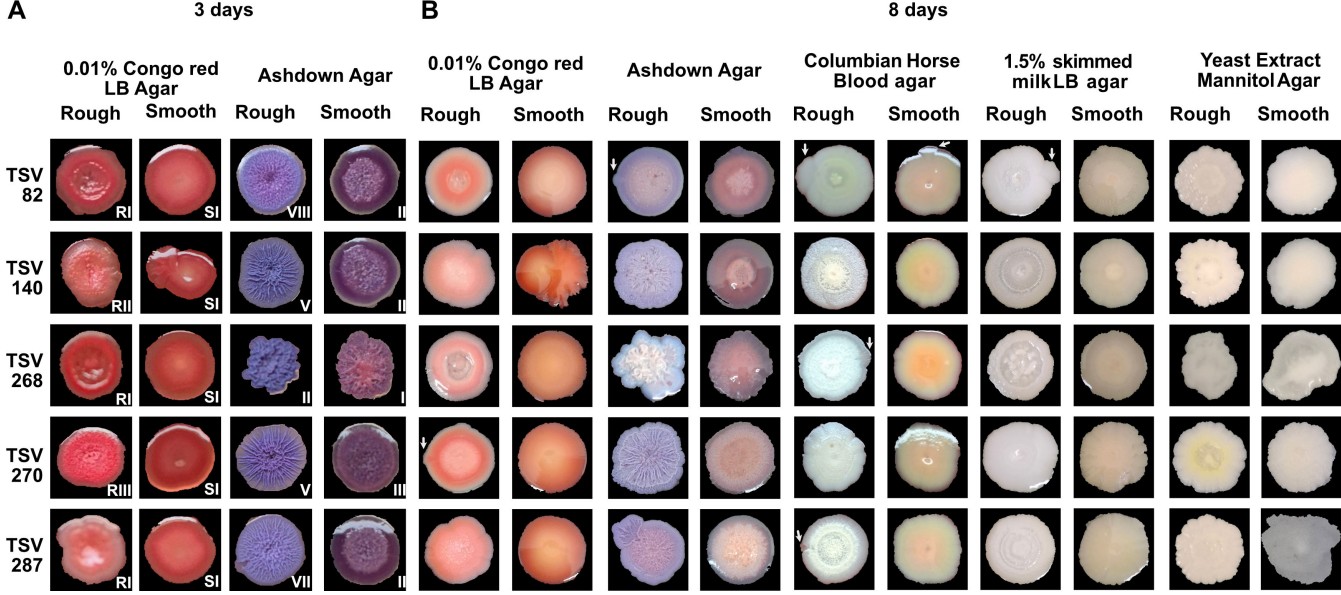

**FIG 1** Characterization of *Bp* isolates by colony morphology and phenotypes. (A) *Bp* isolates are grown on CRLA or Ashdown agar for 3 days. Roman numbers represent the type of colony morphology. (B) Variant emergence among *Bp* isolates when grown on various agar for 8 days; arrows target variants emerging from parental isolates. Note: All rough variants retained a macroscopically wrinkled or dry texture when observed by eye, across media.

further characterization. To confirm morphology differences, the parental strain and its variants were also grown on Ashdown agar (Fig. 1). Interestingly, TSV82, TSV268, and TSV287 smooth variants were consistently classified as type SI on CRLA and type II (TSV82 and TSV287 smooth) and type I (TSV268 smooth) on Ashdown agar, whereas their respective rough CMV forms were further differentiated. On CRLA, TSV82 and TSV268 rough variants were classified as RI, and TSV287 rough as SI. On Ashdown, these rough variants were more distinct, with TSV82 rough classified as type VIII, TSV268 rough as type II, and TSV287 rough as type VII. In contrast, TSV140 and TSV270 smooth variants were classified as SI on CRLA and type II and III on Ashdown, respectively, whereas their rough forms were more divergent: TSV140 rough was RII on CRLA and type V on Ashdown, and TSV270 rough was RIII on CRLA and type V on Ashdown (19, 56). With extended incubation, some morphotypes shifted (Fig. 1B). TSV270 rough on CRLA developed a lighter inner colony, whereas TSV268 smooth lost its wrinkled texture and became fully smooth.

Since CMV often reflects underlying adaptive phenotypes, we next assessed virulence-associated traits, including proteolytic and hemolytic activity, mucoidy, and antibiotic susceptibility (Table 1). All ten rough and smooth CMVs were sensitive to meropenem (MEM) and ceftazidime (CAZ). However, although all ten rough CMVs were sensitive to TMP-SMX, the smooth variant of TSV140 was resistant, whereas the smooth variant of TSV268 remained classified as "Susceptible, Increased Exposure" but was more sensitive than its rough counterpart.

As expected, smooth variants were generally more mucoid on YEM agar than their respective rough parental forms. Additionally, rough variants of TSV82, TSV270, and TSV287 displayed hemolytic activity, in contrast to their smooth counterparts. TSV82 and TSV270 rough forms also produced significantly more protease and siderophore than the smooth forms, whereas TSV287 rough only displayed a higher production of siderophore compared with its smooth variant.

To further explore CMV, we assessed the formation of sectors (bacterial growth on the edges of the main colony), reflecting in variant emergence, after extended incubation on each medium (Fig. 1B). CRLA induced smooth variants in the initial rough forms of TSV268 and TSV270, whereas blood agar promoted smooth variants in the rough forms of TSV82, TSV268, and TSV287 but also in the smooth form of TSV82. Ashdown agar triggered morphotype changes only in TSV82 and TSV287. Finally, TSV82 smooth variants emerged from its initial rough form on LB agar supplemented with skimmed milk. TSV140 showed no variant emergence across any media.

**TABLE 1** Phenotypic profiling of seven Bp clinical isolates and their respective mucoid variants[a,c,d,e,f]

| Isolate | Colony morphology | Antibiotic susceptibility test | | | Mucoidy | Hemolysis | Proteolytic activity (cm$^2$) | Siderophore production (cm$^2$) |
|---------|-------------------|------|-----|---------|---------|-----------|-------------------------------|---------------------------------|
| | | MEM | CAZ | Tmp-smx | | | | |
| TSV 82 | Rough | S | I | I | − | Beta | 1.81 ± 0.51* | 7.473 ± 1.239* |
| | Smooth | S | I | I | ++ | Gamma | 0.89 ± 0.09 | 5.097 ± 0.911 |
| TSV 140 | Rough | S | I | I | + | Gamma | 0.73 ± 0.10 | 4.113 ± 0.728 |
| | Smooth | S | I | R[b] | ++ | Gamma | 0.71 ± 0.00 | 4.634 ± 1.237 |
| TSV 268 | Rough | S | I | I | − | Gamma | 0.00 ± 0.00 | 2.058 ± 0.548 |
| | Smooth | S | I | I[b] | + | Gamma | 0.00 ± 0.00 | 2.307 ± 0.288 |
| TSV 270 | Rough | S | I | I | + | Beta | 1.80 ± 0.23** | 13.888 ± 2.865** |
| | Smooth | S | I | I | ++ | Gamma | 0.64 ± 0.07 | 3.753 ± 1.46 |
| TSV 287 | Rough | S | I | I | − | Beta | 0.75 ± 0.10 | 7.707 ± 0.999* |
| | Smooth | S | I | I | + | Gamma | 0.92 ± 0.13 | 5.6 ± 0.916 |

[a]EUCAST: S = Susceptible; I = Susceptible increased exposure; R = Resistant; Meropenem (MEM): S ≤ 2; R > 2; Ceftazidime (CAZ; disk 10 µg): S ≥ 50 mm; R < 18 mm; Trimethoprim-Sulphamethadoxone (TMP-SMX; disk 1.25–23.75 µg): S ≥ 50 mm; R < 17 mm.
[b]TMP-SMX susceptibility statistically different between rough and smooth CMV of TSV140 and TSV268 using a *t*-test.
[c]Mucoidy: − = non-mucoid; + = mucoid; ++ = more mucoid.
[d]Hemolysis: beta = complete lysis of red blood cell; gamma = no lysis.
[e]Proteolytic activity: average ± standard deviation; comparison TSV 270 rough vs. smooth by t-test and TSV 82 rough vs. smooth by Wilcoxon test; Siderophore production: average ± standard deviation; comparison by t-test.
[f]P values: "*" represents P value < 0.05; "**" represents 0.01 < P value < 0.05.

Taken together, these results demonstrate that *Bp* colony morphology varies across some media but does not reliably correlate with virulence-related phenotypes.

## Genomic analyses reveal distinct mutations, including in the noncoding region, with potential predicted regulatory elements

After confirming that both rough and smooth colony morphotypes from each isolate shared the same sequence type (ST; Table S1), we investigated whether differences in colony morphology were due to underlying genomic variations. We first assembled the genomes of TSV82 rough, TSV140, TSV268, TSV270, and TSV270 smooth colonies and mapped Illumina reads from TSV82 smooth, TSV140, TSV268, TSV270, and TSV270 rough on their respective counterparts' assembled genomes using Snippy (Table 2).

First, no genomic variation was detected between the rough and smooth forms of TSV268 and TSV270, suggesting that their CMV is likely driven by phase variation. In contrast, TSV82 smooth differed from its rough counterpart by 33 mutations, including four in intergenic regions, 16 synonymous, and eleven missenses. To confirm the accuracy of these mutations, we mapped contigs from the TSV82 smooth Illumina assembly onto the TSV82 rough reference to ensure that they were not artifacts associated with contig ends. However, the Illumina coverage was $50.5\times$ for contig 1 and $41.4\times$ for contig 2, which could account for the observed mutations (Table 2).

We then assessed the predicted impact of intergenic and missense mutations based on amino acid substitution significance and genomic context (e.g., within open reading frames or promoter regions). Notable mutations were identified in several proteins: MbaJ, a protein part of the siderophore biosynthesis cluster (which could explain the observed difference in siderophore production for this isolate; Table 1; Fig. S1A); PotF, a periplasmic binding protein of the PotFGHI system involved in putrescine import in *E. coli* (Fig. S1B); TSV82_RS_18625/BPSS0830, a DNA-binding protein; a NAD(P)-dependent oxidoreductase (Fig. S1C); and TSV82_RS_19610/BPSS0120, a helix-turn-helix domain-containing protein and putative member of the AraC family of transcriptional regulators (Fig. S1D). To evaluate the potential structural consequences of these mutations, we used AlphaFold 4 to predict and align the tertiary structures of wild-type and mutated proteins. Mutations were either located in regions with low prediction confidence or did not significantly alter predicted secondary or tertiary structures (Fig. S1). Further experimental validation will be necessary to assess their functional impact. Finally, we identified mutations between smooth and rough colonies within predicted ncRNAs located in the 5′ untranslated regions (UTRs) of TSV82_RS_19610 (a G-to-T substitution), TSV82_RS_11330 (a deletion of "CG"), and TSV82_RS_26200 (a G-to-T substitution).

For TSV140, rough and smooth variants differed by a deletion of one set of "GCTTCG" repeat sequence within TSV140_RS_30990 encoding for a putative serine/arginine repetitive matrix 1 (Table 2). This deletion induces a frame shift, resulting in a putative truncated protein. In TSV287, the smooth variant carried seven repeats of the sequence "ACGAATCAGT" while the rough variant lacked one ($n = 6$) in the intergenic region between the 3′ UTR of *ispA* (TSV287_RS_03530) and 5′ UTR *trnP* tRNA (TSV287_RS_03535) (Table 2).

## Proteomic profiles of *Bp* colony morphotypes reflect adaptive changes in virulence, quorum sensing, and DNA methylation

In this study, we assessed phenotypes associated with key virulence factors known as essential for *Bp* pathogenicity, including protease, siderophore, and EPS production, along with susceptibility to antibiotics commonly used for melioidosis treatment. However, *Bp* is also known to modulate its production of lipopolysaccharide (LPS) and capsular polysaccharide (CPS) to escape host immunity, resistance to antibiotics and phage attack, employ multiple secretion systems to invade within macrophages (to replicate) and then escape (57–67). To gain deeper insights into the molecular pathways affected by CMV, we performed proteomic analyses comparing smooth and rough colony morphologies when grown on CRLA plates for 3 days (Fig. 2; Tables S4 to S10).

**TABLE 2** Genomic modifications between rough and smooth CMV

| Comparison | Average of Illumina coverage | Chromosome | Position | Type | Reference | Alternative | Evidence | Frame type | Effect | Locustag | Gene | Product | *Bp* K9623 locustag |
|---|---|---|---|---|---|---|---|---|---|---|---|---|---|
| TSV82 Smooth vs rough | Contig 1 50.5× Contig 2 41.4× | 1 | 87,085 | SNP | G | A | A:39 G:3 | CDS | Synonymous c.3825G > A p.Thr1275Thr | RS_00350 | *mbaJ* | Siderophore-related no-ribosomal peptide synthase | BPSL1778 |
| | | 1 | 87,236 | SNP | T | C | C:42 T:2 | CDS | Synonymous c.3976T > C p.Leu1326Leu | | | | |
| | | 1 | 87,274 | SNP | C | A | A:40 C:1 | CDS | Missense c.4014C > A p.His1338Gln | | | | |
| | | 1 | 87,454 | SNP | C | T | T:50 C:1 | CDS | Synonymous c.4194C > T p.Ile1398Ile | | | | |
| | | 1 | 88,446 | SNP | C | T | T:16 C:1 | CDS | Missense c.5186C > T p.Ala1729Val | | | | |
| | | 1 | 88,575 | SNP | A | C | C:21 A:2 | CDS | Missense c.5315A > C p.Asp1772Ala | | | | |
| | | 1 | 88,933 | SNP | C | T | T:16 C:1 | CDS | Synonymous c.5673C > T p.Arg1891Arg | | | | |
| | | 1 | 89,173 | SNP | G | A | A:24 G:2 | CDS | Synonymous_ c.5913G > A p.Glu1971Glu | | | | |
| | | 1 | 89,512 | SNP | C | A | A:20 C:1 | CDS | Synonymous c.6252C > A p.Ala2084Ala | | | | |
| | | 1 | 153,859 | SNP | T | A | A:28 T:2 | CDS | Missense c.167A > T p.His56Leu | RS_00580 | | AMP-binding domain protein | BPSL1734 |
| | | 1 | 991,823 | SNP | A | T | T:13 A:1 | CDS | Missense c.217A > T p.Thr73Ser | RS_04110 | | Oxidoreductase, FAD-binding family protein | BPSL2405 |
| | | 1 | 2,552,774 | Del | CG | C | C:54 CG:4 | Intergenic region | | | | Putative non-coding region with regulatory element in 5' RS_11330 | BPSL0367 |
| | | 1 | 2,942,569 | SNP | G | A | A:10 G:0 | CDS | Synonymous c.636G > A p.Ala212Ala | RS_12950 | | Phosphoribosyl transferase | BPSL0718 |
| | | 1 | 2,942,620 | SNP | C | G | G:10 C:0 | CDS | Synonymous c.687C > G p.Ala229Ala | | | | |
| | | 1 | 2,942,657 | SNP | G | A | A:15 G:0 | | | | | | | |
| | | 2 | 262,512 | SNP | T | C | C:27 T:2 | CDS | Missense c.628A > G p.Lys210Glu | RS_18625 | | DNA-binding protein | BPSS0830 |
| | | 2 | 486,060 | SNP | G | A | A:28 G:1 | CDS | Missense c.601G > A p.Glu201Lys | RS_19540 | | SDR family NAD(*P*)-dependent oxidoreductase | BPSS1007 |

TABLE 2 Genomic modifications between rough and smooth CMV (Continued)

| Comparison | Average of Illumina coverage | Chromosome | Position | Type | Reference | Alternative | Evidence | Frame type | Effect | Locustag | Gene | Product | Bp K9623 locustag |
|---|---|---|---|---|---|---|---|---|---|---|---|---|---|
| | | 2 | 537,160 | Complex | GGCCGG | AATCGA | AATCGA:10 GGCCGG:0 | CDS | Missense c.869_874delCCGGCCinsTC-GATT p.ProGly-Pro290LeuAspSer | RS_19610 | | Helix-turn-helix domain protein putative AraC family members | BPSS1020 |
| | | 2 | 538,080 | SNP | G | T | T:16 G:1 | Intergenic region | | | | Putative non-coding region with regulatory element in 5' RS_19610 | |
| | | 2 | 547,475 | SNP | C | A | A:16 C:1 | CDS | Synonymous c.285G > T p.Ala95Ala | RS_19655 | | Branched-chain amino acid ABC transporter permease | BPSS1030 |
| | | 2 | 685,277 | SNP | G | A | A:29 G:1 | CDS | Missense c.194C > T p.Ala65Val | RS_20305 | hyfB | hydrogenase four subunit B | BPSS1147 |
| | | 2 | 737,147 | SNP | A | G | G:30 A:2 | CDS | | | | | |
| | | 2 | 746,638 | SNP | A | G | G:30 A:0 | CDS | Synonymous c.783T > C p.Ser261Ser | RS_20510 | | Luciferase oxidoreductase, group one family protein | BPSS1190 |
| | | 2 | 988,639 | SNP | A | G | G:55 A:4 | CDS | Missense c.164A > G p.Gln55Arg | RS_21370 | | Peptidase, S9A (Prolyl oligopeptidase) family | BPSS1346 |
| | | 2 | 1,313,262 | SNP | C | T | T:12 C:0 | CDS | Synonymous c.207C > T p.Ile69Ile | RS_22655 | | PPC domain-containing protein | BPSS1588 |
| | | 2 | 1791476 | SNP | C | T | T:22 C:1 | CDS | Synonymous c.2052C > T p.Ala684Ala | RS_24525 | rbbA | ribosome-associated ATPase/putative transporter RbbA | BPSS1938 |
| | | 2 | 2,200,644 | SNP | G | T | T:20 G:1 | Intergenic region | | | | Putative non-coding region with regulatory element in 5' RS_26200 | BPSS2260 |
| | | 2 | 3,013,633 | SNP | G | C | C:69 G:0 | CDS | Synonymous c.30C > G p.Ala10Ala | RS_29435 | potI | ABC-type spermidine/putrescine transport system, permease component II | BPSS0464 |
| | | 2 | 3,015,293 | SNP | A | G | G:38 A:2 | CDS | Synonymous c.492T > C p.Arg164Arg | RS_29445 | potG | Spermidine/putrescine import ATP-binding protein | BPSS0466 |
| | | 2 | 3,015,662 | SNP | A | G | G:37 A:1 | CDS | Synonymous. 123T > C p.Ala41Ala | | | | |
| | | 2 | 3,016,583 | SNP | G | C | C:53 G:4 | CDS | Synonymous c.513C > G p.Ala171Ala | RS_29450 | potF | Putrescine-binding periplasmic protein | BPSS0467 |
| | | 2 | 3,016,945 | SNP | G | A | A:47 G:1 | CDS | Missense c.151C > T p.Pro51Ser | | | | |
| | | 2 | 3,017,125 | SNP | G | A | A:24 G:0 | promoter | | | | | |

**TABLE 2** Genomic modifications between rough and smooth CMV (*Continued*)

| Comparison | Average of Illumina coverage | Chromosome | Position | Type | Reference | Alternative | Evidence | Frame type | Effect | Locustag | Gene | Product | *Bp* K9623 locustag |
|---|---|---|---|---|---|---|---|---|---|---|---|---|---|
| TSV140 Rough vs smooth | Contig 1 67.8x Contig 2 69.8x Contig 3 90x | 2 | 2,170,684 | Del | AGCTTCG | A | A:10 AGCTTCG:0 | CDS | Missense c.53_58delAGCTTCG p.PheGly103Phe | RS_30990 | | serine/arginine repetitive matrix 1 CDS | BPSS1650 |
| TSV268 Rough vs smooth | Contig 1 55.7x Contig 2 56.7x | | | | | | | | _a | | | | |
| TSV270 Rough vs smooth | Contig 1 74x Contig 2 76.3x | | | | | | | | — | | | | |
| TSV287 Rough vs smooth | Contig 1 64.6x Contig 2 55.4x | 1 | 774,690 | Del | GACGAATC AG GT | G:32 GACGAATC AGT:0 | | Intergenic region | | RS_03535 | | Putative non-coding region with regulatory element between 3′ ispA (RS_03530) and 5′ trnP tRNA (RS_03540) | BPSL3006- BPSL3007 |

*a*—, not applicable.

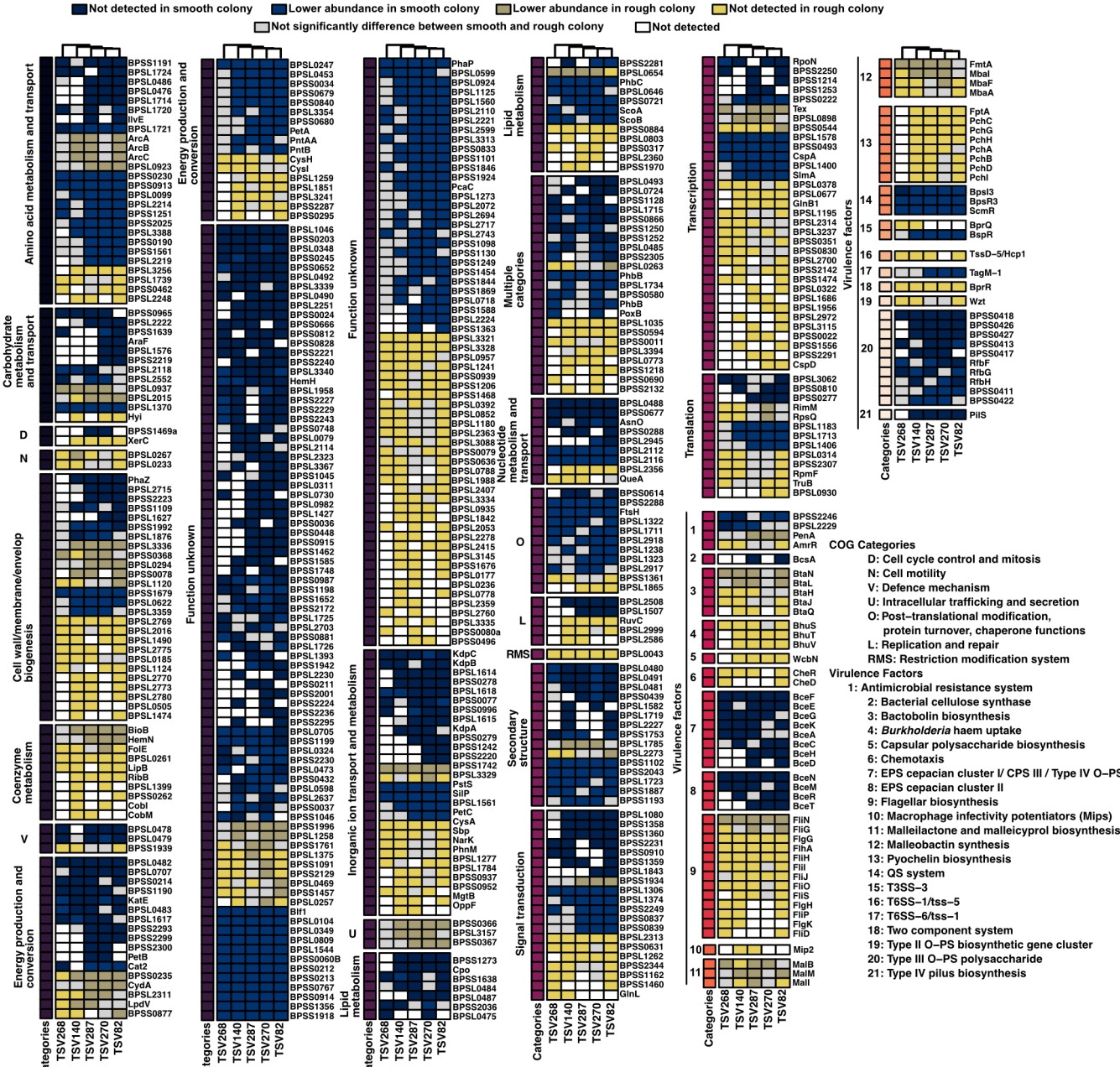

**FIG 2** Proteomic profile of *Bp* isolates by comparing smooth to rough colonies. Heatmap showing protein abundance difference, for each *Bp* isolates for common proteins between isolates.

Differential abundance was observed in proteins involved in a wide range of biological functions including amino acid and carbohydrate metabolism and transport, cell cycle control and division, cell wall/membrane/envelope biogenesis, defense mechanisms, energy production and conversion, inorganic ion transport and metabolism, lipid metabolism, nucleotide metabolism, post-translational modification and chaperone function, secondary structure, signal transduction, transcription, and translation. These proteins were consistently either undetected or significantly lower or higher in abundance in at least three smooth colonies compared with their rough counterparts. Notably, proteins associated with cell motility and intracellular trafficking and secretion were typically undetected or present in lower abundance in rough colonies relative to smooth colonies.

More specifically, proteins involved in the CAC$^6$AG DNA methylation motif (Tables S2 and S3), antimicrobial resistance (PenA, AmrR), bactobolin biosynthesis (BtaN, L, H, J, Q), heme uptake (BhuS, T, V), CPS biosynthesis (WbcN), chemotaxis (CheR, CheD), flagellar biosynthesis (FliN, G, FlgG, H, K, FlhA, FliD, H, I, J, O, S, P), macrophage infectivity potentiators (Mip2), malleilactone and malleicyprol biosynthesis (MalB, I, M), malleobactin synthesis (FmtA, Mbal, F, A), pyochelin biosynthesis (FptA, PchA, B, C, D, G, H, I), T3SS-3 secretion system (BprQ), T6SS-1 secretion system (TssD-5 aka Hcp1), two-component signal transduction system (BprR), and Type II-OPS biosynthesis (Wzt) are not detected or in lower abundance in rough colonies relative to smooth colonies. In contrast, proteins involved in bacterial cellulose synthase (BcsA), EPS cepacian cluster I and II (BceA, C, D, E, F, G, H, N, M, R, and T), quorum sensing (BpsI3/R3 and ScmR), T3SS-3 cascade regulator BspR, T6SS-6 (TagM-1, lipoprotein), Type III O–PS polysaccharide (e.g., RbfbF, G, H), and Type IV pilus biosynthesis (PilS), were not detected or in lower abundance in smooth colony when compared with rough colonies (Fig. 2; Tables S4 to S10).

Altogether, these results indicate that differential regulation of processes such as quorum sensing and DNA methylation leads to distinct production patterns of numerous proteins involved in virulence, which in turn are reflected in the observed CMV phenotypes. Although these findings highlight that CMV reflects underlying adaptive changes linked to virulence modulation, further studies are required to determine whether ScmR, quorum sensing, and DNA methylation directly regulate CMV and associated modulation of virulence factor production.

## Mutations in non-coding regulatory elements may contribute to differential protein abundance but are unlikely to drive CMV

Given the identification of mutations in non-coding intergenic regions that may contain regulatory elements such as small (s)RNA (with their predicted structure in Fig. S2), we investigated whether these regulatory elements could account for the observed differential protein expression (Table S11). First, none of the potential predicted ncRNAs had been previously reported (68, 69). Target prediction linked these predicted ncRNAs to genes involved in Type III (*bsaV, bsaK*) and Type IV (*vgrG, bimA*) secretion systems, antimicrobial resistance (*penA*), capsular polysaccharide biosynthesis (*wcbK*), chemotaxis (*cheY*), and secondary metabolite biosynthesis (e.g., *pchD/E* for pyochelin, *malF/M* for malleilactone, *mbaE* for malleobactin, *bceQ* for EPS cepacian, and *bhuU* for Burkholderia heme uptake).

However, most of these targets did not exhibit significant differences in protein abundance between rough and smooth colonies in our proteomic analyses. Thus, although the non-coding regions merit further investigation for potential regulatory elements, they do not appear to be the primary drivers of colony morphotype variation.

## Deeper omics profiling shows potential to better characterize *Bp* isolates

Identification based solely on phenotype is unreliable due to the inherent variation in *Burkholderia* colony morphotypes (19, 56), which is further complicated by CMV occurring within individual isolates as an adaptive response. Our combined phenotypic and genomic analyses revealed that these genomic modifications do not alone explain the observed changes (Fig. 1; Table 2). This was confirmed by principal component analysis (PCA), where phenotypic and genomic data failed to cluster similar morphotypes or isolates (Fig. 3A). In contrast, proteomic profiling allowed more consistent discrimination between rough and smooth CMV variants and even between distinct smooth variants (Fig. 3A).

However, proteomics alone lacks biological context without accompanying phenotypic data, particularly for virulence traits, pathogenicity, and responses to antimicrobial treatment or stress, which may also be shaped by spontaneous mutations. Therefore, integrating all data sets provides the most biologically relevant insights (Fig. 3B). Although smooth isolates are not further distinct by integrating phenotypic and genomic data, further rearrangements are observed between rough isolates.

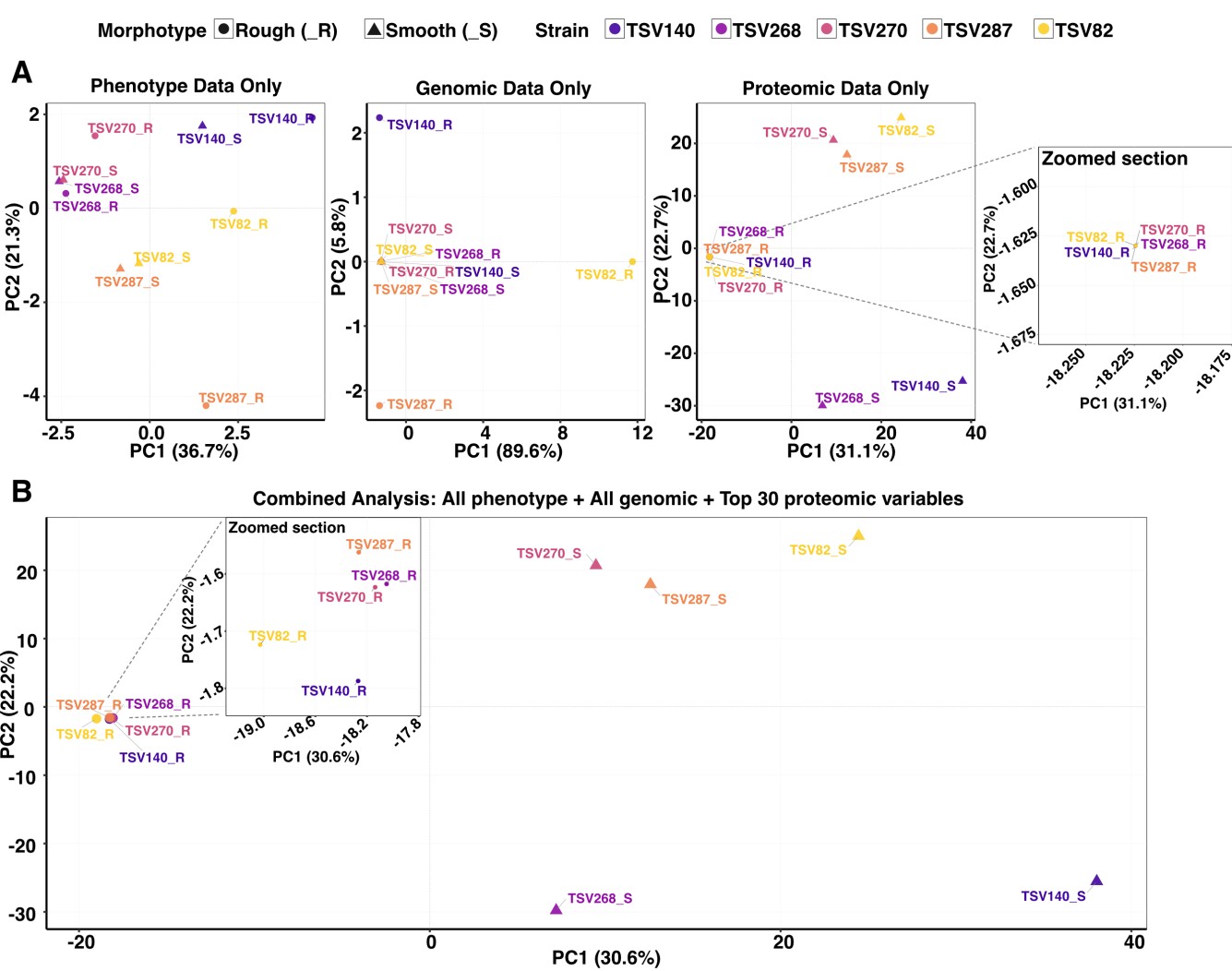

**FIG 3** Cluster of *Bp* isolates and their variants based on three omics layers. (A) Individual PCA analysis based on either phenotype (16), genomic (19), or proteomic (1,429) variables. (B) Integrative PCA analysis using all genomic and phenotype variables and the top 30 proteins differentiating isolates.

## DISCUSSION

*Bp* has been previously reported to exhibit a heterogeneous population, with both mucoid and non-mucoid colonies, and to undergo CMV depending on the presence of environmental pressures. Although genomic profiling is often used to study *Bp* adaptation during infection, and phenotypic assays are used to characterize CMV, integrated studies examining the relationship between genomic changes, phenotypic shifts, and proteomic alterations remain rare (26, 27, 70). In this study, we employed a multi-omics approach to investigate phenotypic and proteomic changes associated with CMV, along with underlying genomic modifications. Our aim was to reveal both conserved and isolate-specific features associated with the switch between rough and smooth morphotypes, which might transition from an acute-like to a chronic-adapted state during infection (63).

In several isolates, phase variation appeared to drive the switch between rough and smooth colony morphologies in the absence of detectable genomic variation. In contrast, other isolates exhibited limited but notable genomic modifications. In TSV82, SNPs led to missense mutations in a protein involved in siderophore biosynthesis, along with changes in both the promoter region and ORF of *potF*, a gene encoding a potential protective antigen for melioidosis (71). However, it is important to note that the genomic

variation observed in TSV82 may be partially attributed to lower Illumina read coverage (below 50×). Additionally, across TSV82 and TSV287 isolates, either a missing repeat sequence or an SNP was detected within a putative novel mRNA regulatory element (68, 69).

Predicted targets of the putative regulatory element suggest involvement in macrophage escape, giant multinucleated cell formation, possibly through predicted interaction with genes involved in type III and IV secretion systems, and the regulation of several biosynthetic gene clusters linked to virulence and antimicrobial resistance, including those responsible for malleilactone, malleicyprol, malleobactin, and pyochelin biosynthesis. Although some key regulators, such as BspR, known to regulate the T3SS-3 and influence T6SS-5, showed differential abundance in the proteomics analysis, other secretion system components remained unchanged. This suggests that further studies are needed to clarify the regulatory changes underlying CMV that influence both secretion systems, which are critical for macrophage invasion and intracellular survival during infection (57–66).

Nonetheless, our data show extensive changes in pathways related to flagella, antimicrobial molecules, siderophore biosynthesis, CPS, LPS, and OMPs involved in antimicrobial resistance. These findings support previous studies describing the emergence of SCVs in *Bp* through phase variation, as well as the role of genomic modifications in adaptation during chronic infection (17, 26, 72, 73). The limited number of point mutations observed suggests that although they may contribute to phenotypic changes, they are unlikely to be the primary drivers of CMV. If genomic modifications were the main cause, we would expect more extensive alterations across regulatory genes and isolates.

Instead, the broad phenotypic and molecular changes underlying CMV are more likely driven by global regulatory systems—such as ScmR, quorum sensing, and epigenetic mechanisms like DNA methylation. These pathways were affected at the protein level in our study. This aligns with previous work demonstrating that ScmR regulates virulence factor production and quorum sensing in *B. thailandensis*, a surrogate model of *Bp*, and that both quorum sensing and DNA methylation influence phase variation in *B. ambifaria*, a member of the *Burkholderia cepacia* complex (20, 74, 75).

This study aimed to serve as a preliminary investigation, ultimately demonstrating that integrating multi-omics data—phenotypic, genomic, and proteomic—provides valuable insights into the biological impact reflected by *Bp* adaptation through CMV. This approach allowed us to differentiate isolates by morphotype and to uncover mechanisms underlying bacterial adaptation. It can also resolve *Bp* polyclonal infections that are otherwise mistaken for smooth and rough CMV when assessed by phenotype alone or in combination with single-colony sequencing, as we have recently reported (76). However, limitations remain, particularly in proteomic analyses, where LC-MS/MS detection thresholds may restrict peptide identification. Future work will involve expanding this approach to a broader collection of isolates from environmental, animal, and human sources, spanning acute and chronic infections. We aim to generate a comprehensive database that integrates multi-layered omics data, including epigenomics, transcriptomics, and metabolomics, with virulence phenotypes (e.g., macrophage invasion and escape), antibiotic responses, and clinical outcomes. This will improve our ability to discover the various mechanisms underlying bacterial adaptation, visible through CMV, correlate molecular features with disease severity and persistence, and identify stable biomarkers for diagnostic and therapeutic targets.

## ACKNOWLEDGMENTS

The authors thank Julien L. Breton-Robin for helping set command linesline tools using singularity.

This work was supported by the funding AIMI and UTS Faculty of Science seed fundings held by Dr. Pauline M. L. Coulon, the strategic research accelerator funding (SRA 2726229) held by Prof. Garry Myers and EL2 Investigator grant from the NHMRC

(APP2033851) held by Dr. Patrick N. A. Harris. Part of this work was carried out during the award of a Chancellor's Research Fellowship to Daniel G. Mediati. Royal Australasian College of Physicians Queensland Regional Committee Research Development Grant held by Ian Gassiep allowed the collection and storage of Bp isolates. Experimental work was done at UQCCR which would not have been possible without the ISME scholar funding awarded to Dr. Pauline M. L. Coulon.

## AUTHOR AFFILIATIONS

[1]Faculty of Science, University of Technology Sydney, Australian Institute for Microbiology and Infection, Ultimo, NSW, Australia

[2]Faculty of Medicine, The University of Queensland, The University of Queensland Centre for Clinical Research (UQCCR), Brisbane, Queensland, Australia

[3]Mass Spectrometry Facility, The University of Queensland, Centre for Clinical Research, Brisbane, Queensland, Australia

## AUTHOR ORCIDs

Pauline M. L. Coulon  http://orcid.org/0000-0001-9482-2360
Kay Ramsay  http://orcid.org/0000-0002-6682-9351
Miranda E. Pitt  http://orcid.org/0000-0002-8255-4036
Joyce To  http://orcid.org/0000-0003-3482-4369
Daniel G. Mediati  http://orcid.org/0000-0003-3599-5119
Ian Gassiep  http://orcid.org/0000-0003-3548-6963
Sarah Reed  http://orcid.org/0000-0001-5219-9590
Patrick N. A. Harris  http://orcid.org/0000-0002-2895-0345
Garry S. A. Myers  http://orcid.org/0000-0002-4756-4810

## AUTHOR CONTRIBUTIONS

Pauline M. L. Coulon, Conceptualization, Formal analysis, Funding acquisition, Investigation, Methodology, Visualization, Writing – original draft, Writing – review and editing | Kay Ramsay, Investigation, Writing – review and editing | Aven Lee, Investigation, Writing – review and editing | Edita Ritmejeryte, Investigation, Writing – review and editing | Miranda E. Pitt, Investigation, Writing – review and editing | Joyce To, Investigation | Daniel G. Mediati, Investigation, Writing – review and editing | Ian Gassiep, Funding acquisition, Writing – review and editing | Sarah Reed, Investigation, Writing – review and editing | Patrick N. A. Harris, Funding acquisition, Writing – review and editing | Garry S. A. Myers, Funding acquisition

## DATA AVAILABILITY

Genomes from this study are available in GenBank under the Bioproject: PRJNA1295092 (77, 78). Proteomic raw data deposited to the ProteomeXchange Consortium via the PRIDE partner repository with the data set identifier PXD065768 (79).

## ETHICS APPROVAL

This study received ethical approval from the Royal Brisbane and Women's Hospital Ethics Committee (LNR/2020/QRBW/65573), with site-specific authority obtained from the Townsville Hospital and Health Service and approval under the Queensland Public Health Act.

## ADDITIONAL FILES

The following material is available online.

## Supplemental Material

**Supplemental figures (Spectrum02437-25-s0001.docx).** Fig. S1 and S2.
**Supplemental tables (Spectrum02437-25-s0002.xlsx).** Tables S1 to S11.

## Open Peer Review

**PEER REVIEW HISTORY (review-history.pdf).** An accounting of the reviewer comments and feedback.

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
