## [Reviewer comments · Microbiology Spectrum]

Microbiology Spectrum

Profiling of *Burkholderia pseudomallei* variants derived from Queensland's clinical isolates

Pauline Coulon, Kay Ramsay, Aven Lee, Edita Ritmejerite, Miranda Pitt, Joyce To, Daniel Mediati, Sarah Reed, Patrick Harris, and Garry Myers

Corresponding Author(s): Pauline Coulon, University of Technology Sydney

Review Timeline:

Submission Date:	August 6, 2025
Editorial Decision:	September 14, 2025
Revision Received:	October 3, 2025
Accepted:	November 10, 2025

Editor: Silvia Cardona

Reviewer(s): The reviewers have opted to remain anonymous.

Transaction Report:

DOI: <https://doi.org/10.1128/spectrum.02437-25>

Re: Spectrum02437-25 (Profiling of *Burkholderia pseudomallei* variants derived from Queensland's clinical isolates)

Dear Dr. Pauline M.L. Coulon:

Thank you for submitting your manuscript to Microbiology Spectrum. I see it as a nice example of a multi-omics approach. Two experts reviewed your article and provided comments to improve the quality of your work. I also have a comment regarding the concept of phase variation, which is mentioned several times in the text. Please ensure phase variation is defined with corresponding references.

Reviewers' recommendations are provided below.

Revision Guidelines

Sincerely,
Silvia Cardona
Editor
Microbiology Spectrum

Reviewer #1 (Comments for the Author):

General remark.

The authors confuse cause and effect.

Causality in brief - Modulation of regulatory processes leads to differential gene expression => different set of proteins => different properties, including the formation of different colony morphotypes. In the authors' presentation, it looks the other way around. Phrases like "CMV significantly affects the expression of numerous proteins", "the importance of CMV in modulating virulence", "the biological impact of CMV", and "the impact of CMV on both secretion systems" need to be edited.

L27 The manuscript provides data on the characteristics of smooth and rough morphological variants of colonies. Smooth and mucoid colonies are not the same thing. There are no mucoid colonies in the photographs provided by the authors.

L275 As expected, phenotypic profile do not cluster by colony morphotype - it is an unfortunate phrase in this context.

L 326 ...the formation of sectors - it's not clear.

L134, 319, and 334 - The methods of statistical analysis are not specified; the results do not include the correlation coefficient and p-value.

L377 Proteomic profile of Bp isolates and their CMVs reveal alterations in proteins - This is not consistent with the text that describes the modulation in the synthesis of certain proteins.

Reviewer #2 (Comments for the Author):

Thanks to the authors for their work and the usage of multi-omics to unravel CMV in Bp.

However, there are some points the authors might consider:

The authors could provide a definition of the morphotypes they used. Moreover, data on morphotype switching percentages or morphotype stability (on agar or after liquid culture) is missing. A comparison to BP K96243 would be helpful.

L:85: Did the authors mean antibiotic production or resistance?

L120: A precise number should be provided

L194: The authors should provide the incubation conditions for the experiment.

Altogether, the work displays a basis for future research as stated by the authors in the discussion.

Reviewed Comments addressed

First, we thank the editor and reviewers for taking the time to evaluate our manuscript and for their constructive comments. As requested, we have revised the manuscript and added a definition of phase variation in the Introduction. Below, we provide detailed responses to each individual comment.

Editor's comment:

Please ensure phase variation is defined with corresponding references.

We have added a definition of phase variation in the introduction (line 87).

Reviewer #1 (Comments for the Author):

General remark.

The authors confuse cause and effect.

Causality in brief - Modulation of regulatory processes leads to differential gene expression => different set of proteins => different properties, including the formation of different colony morphotypes. In the authors' presentation, it looks the other way around. Phrases like "CMV significantly affects the expression of numerous proteins", "the importance of CMV in modulating virulence", "the biological impact of CMV", and "the impact of CMV on both secretion systems" need to be edited.

Response: Thank you for highlighting this issue. We have revised all relevant phrases to ensure that CMVs reflect changes in molecular pathways throughout the manuscript. Changes can be found at lines 390, 417, 484, and 506.

L27 The manuscript provides data on the characteristics of smooth and rough morphological variants of colonies. Smooth and mucoid colonies are not the same thing. There are no mucoid colonies in the photographs provided by the authors.

Response: Mucoid and non-mucoid are the major phenotype descriptors for Bp colony morphotypes in the literature. The smooth appearance of colonies reflects variation in EPS production, giving a mucoid-like aspect. Nevertheless, we agree with the reviewer and have changed line 27 to "rough and smooth colony morphologies."

L275 As expected, phenotypic profile do not cluster by colony morphotype - it is an unfortunate phrase in this context.

Response: We have revised the section title to: "Colony morphotype variation does not translate into distinct phenotypic clusters" (line 280).

L 326 ...the formation of sectors - it's not clear.

Response: We have clarified this in the text: “we assessed the formation of sectors (bacterial growth on the edges of the main colony)” (line 332).

L134, 319, and 334 - The methods of statistical analysis are not specified; the results do not include the correlation coefficient and p-value.

Response: We have added a subsection under phenotypic analyses: “Statistical analyses. All datasets were first tested for normality using the Shapiro–Wilk test. Based on the results, comparisons of virulence factor production between rough and smooth CMVs were performed using either a Wilcoxon rank-sum test or a t-test, with Bonferroni-adjusted p-values.” (Lines 145–138)

L377 Proteomic profile of Bp isolates and their CMVs reveal alterations in proteins - This is not consistent with the text that describes the modulation in the synthesis of certain proteins.

Response: We have revised the section title to: “Proteomic profiles of Bp colony morphotypes reflect adaptive changes in virulence, quorum sensing, and DNA methylation” (line 383).

Reviewer #2 (Comments for the Author):

Thanks to the authors for their work and the usage of multi-omics to unravel CMV in Bp.

However, there are some points the authors might consider:

The authors could provide a definition of the morphotypes they used. Moreover, data on morphotype switching percentages or morphotype stability (on agar or after liquid culture) is missing. A comparison to BP K96243 would be helpful.

Response: We thank the reviewer for this suggestion. We have added a definition distinguishing CMV and phase variation on line 34: CMV comprises both irreversible and reversible changes, whereas phase variation refers to reversible, dynamic switches between colony morphotypes. Based on the literature, phase variation occurs at a frequency of $\sim 10^{-5}$, whereas mutations occur at $\sim 10^{-8}$ (Wisniewski-Dyé and Vial, 2008; DOI: 10.1007/s10482-008-9267-0). In our study, we confirmed morphotype variation by streaking isolates on different solid media and observing consistent phenotypes. While we did not quantify the frequency of occurrence of the different CMVs, our focus was on the molecular and proteomic characterization of morphotypes, and whole-genome sequencing provides more accurate and informative insights into the genetic differences underlying these phenotypes than frequency-based measurements. Relevant comparisons to Bp K96243 have been added where appropriate.

L:85: Did the authors mean antibiotic production or resistance?

Response: Thank you for catching this mistake. We have corrected this to “antibiotic resistance” (line 85).

L120: A precise number should be provided

Response: We have specified the total number of isolates collected from Townsville. To respect the unpublished nature of Etipola and colleagues' work and current authorship, we have modified the sentence to avoid revealing unpublished results (line 121).

L194: The authors should provide the incubation conditions for the experiment.

Response: In the "Protein extraction and preparation" subsection, we have added growth conditions:

"After three days on CRLA incubated at 37°C, bacterial colonies were resuspended in lysing buffer (1 M Tris-HCl containing 1% SDS), boiled at 95°C, and cooled for 10 min at 4°C, repeated three times. Bacterial lysates were then..." (line 197).

Altogether, the work displays a basis for future research as stated by the authors in the discussion.

Re: Spectrum02437-25R1 (Profiling of *Burkholderia pseudomallei* variants derived from Queensland's clinical isolates)

Dear Dr. Pauline M.L. Coulon:

My apologies for the delay in getting a response from the reviewers. I am happy to inform you that your manuscript has been accepted and I am forwarding it to the ASM production staff for publication. Your paper will first be checked to make sure all elements meet the technical requirements. ASM staff will contact you if anything needs to be revised before copyediting and production can begin. Otherwise, you will be notified when your proofs are ready to be viewed.

Sincerely,
Silvia Cardona
Editor
Microbiology Spectrum

Reviewer #1 (Comments for the Author):

The integrated approach to the problem of colony morphotype variation commands sincere respect. I believe that good work deserves a proper presentation, eliminating ambiguity. In my first comment, I would like to draw the authors' attention to the accuracy of their wording. Perhaps this is due to translation difficulties from Australian English, but the phrases "the molecular pathways impacted by CMV" and "the broad phenotypic and molecular changes associated with CMV are more likely driven by global regulatory systems..." have opposite meanings for a general audience.

Reference 20 (L604) contains an unfortunate error.